# Relative Humidity and Its Impact on the Immune System and Infections

**DOI:** 10.3390/ijms24119456

**Published:** 2023-05-29

**Authors:** Gabriella Guarnieri, Bianca Olivieri, Gianenrico Senna, Andrea Vianello

**Affiliations:** 1Department of Cardiac-Thoracic-Vascular Sciences and Public Health, Respiratory Pathophysiology Division, University of Padova, 35128 Padova, Italy; andrea.vianello.1@unipd.it; 2Department of Medicine, University of Verona, 37134 Verona, Italy; biancaolivieri92@gmail.com (B.O.); gianenrico.senna@univr.it (G.S.)

**Keywords:** molecular mechanisms, prevention, asthma, allergic rhinitis, infection, respiratory diseases, risk factors

## Abstract

Relative humidity (RH) represents an underestimated outdoor and indoor environmental parameter. Conditions below and above the optimal range could facilitate infectious transmission as well as the exacerbation of respiratory diseases. The aim of this review is to outline the consequences for health of suboptimal RH in the environment and how to limit this negative impact. RH primarily affects the rheological properties of the mucus, modifying its osmolarity and thus the mucociliary clearance. The integrity of the physical barrier, maintained by mucus and tight junctions, is critical for protection from pathogens or irritants. Moreover, the control of RH seems to be a strategy to prevent and control the spread of viruses and bacteria. However, the imbalance of RH in the outdoor and indoor environments is frequently associated with the presence of other irritants, allergens, and pathogens, and therefore the burden of a single risk factor is not clearly defined in different situations. Nonetheless, RHmay have a synergistic negative effect with these risk factors, and its normalization, if possible, may have a positive impact on a healthier environment.

## 1. Introduction

Relative humidity (RH) is a crucial environmental parameter that is often overlooked in discussions on indoor and outdoor air quality. RH is defined as the amount of water vapor present in the air, expressed as a percentage of the amount needed for saturation at the same temperature. RH plays a critical role in maintaining healthy indoor environments by regulating moisture levels and preventing the growth of harmful microorganisms. Emerging evidence suggests it plays a crucial role in human health, particularly in relation to respiratory conditions and allergies. While RH has historically received less attention than other environmental factors, such as pollution, recent studies have emphasized its significance.

Choosing the appropriate range of RH can be challenging, as it is not only affected by temperature but also by the conflicting health effects of an increase or decrease in relative humidity. Conditions that fall outside of the optimal range of 40–60% can have significant impacts on health, including facilitating infectious transmission and exacerbating respiratory diseases (Figure 1) [1]. When RH is too low, it can cause dryness and irritation of the respiratory tract and skin, making individuals more susceptible to infections. On the other hand, when RH is too high, it can create a damp environment that encourages the growth of harmful microorganisms like mould, bacteria, and viruses. Bacteria such as *Staphylococcus Aureus*, *Pseudomonas Aeruginosa*, *Enterobacter* species, and *Legionella* species almost always contaminate humidification and air-conditioning equipment, and their growth is below 30% and above 60% RH. Likewise, influenza and para-influenza viruses, myxovirus, poliovirus, adenovirus increase at a RH below 50% and above 70%. In winter, no mites were found at RH below 50%, while the prevalence of indoor fungi in the indoor environment is relevant above 60% RH. The effects of high RH increase the off gassing of formaldehyde from building and furnishing materials, and the release of odors, microparticles and vapors such as acrolein. Low RH promotes the formation of indoor ozone, which has an irritating effect on the mucous membrane, eyes, nose, throat, and respiratory tract. Different mechanisms account for the damage due to a RH out of the normal range. The main drawback is a negative impact on mucociliary clearance and consequently on the epithelial integrity of the respiratory and conjunctival mucosa. Finally, HR in the atmosphere may favour pollution and have an impact on pollen production [1,2,3]. The climate changes are the result of a complex interaction between climatological factors and pollutants, in which the role of a single element, e.g., RH, cannot be easily identified. Nevertheless, while the variation of HR in the atmosphere may predict increased allergen load or the risk of thunderstorms, which are risk situations for asthmatic patients, in the indoor environment, normalization of HR may have a positive impact on health by reducing irritants and the load of mites and moulds. 

Understanding the impact of RH on respiratory health, allergies, and the immune system is necessary for developing effective preventive measures and public health policies.

The aim of this review is to provide a comprehensive overview of the topic, outline the health consequences from suboptimal RH in the environment, and show how to limit this negative impact.

## 2. Mechanisms

RH primarily affects the rheologic properties of the mucus, modifying its osmolarity and consequently the mucociliary clearance (Table 1). In fact, in the lower airways, the presence of an excess of mucus or its increased thickness may promote bacterial engraftment, especially in the presence of genetic or acquired alterations (chronic obstructive pulmonary disease (COPD), bronchiectasis). On the other hand, in the upper airways, nasal breathing of cold air increases the viscosity of the mucus and reduces the ciliary beat, leading to an increased susceptibility to viral infections. Moreover, the alteration of the mucosal layer can lead to a disruption of the epithelial junctions, thus increasing the permeability to pollutants, allergens, and viruses. Notably, the damaged epithelial cells may release mediators such as alarmins (IL33, TSLP, and IL25), which can promote the inflammatory cascade (Figure 2). On the other hand, cytokines such as IL-4 and IL-13, released in the type 2 inflammatory response, downregulate tight junction barrier protein expression, contributing to reduced epithelial junctional integrity and increased barrier permeability in a vicious circle [2,3,4].

The integrity of the physical barrier, maintained by mucus and tight junctions, is critical for protection from pathogens or irritants (Table 1, Figure 2). The innate immune cells represent the first line of immune barrier. Epithelial dysfunction is also associated with chronic epithelial-to-mesenchymal transition and basement membrane thickening [3] and is related to different factors such as genetics, allergens, injury, infection, autoimmune defects, etc. Moreover, it can be exacerbated by type 2 inflammation [3,4]. Patients with humoral immune deficiency (e.g., B-cell immunodeficiencies) may have an increased risk of barrier dysfunction, due to the overproduction of antibodies, which can lead to inflammation and an inadequate protective humoral immune response against microbes [10].

Alterations of the mucus layer are particularly relevant in patients with preexisting rare diseases which per se affect the muco-ciliary clearance, as cystic fibrosis, ciliary dysfunction or more common conditions as COPD.

In cystic fibrosis, an autosomal recessive genetic disease linked to a mutation in chromosome 7 of the gene encoding a protein that controls the chlorine channel (CFTR: Cystic Fibrosis Transmembrane Conductance Regulator), thick and viscous mucus is produced, involving several organs such as the lung, pancreas, and liver, and is also responsible for infertility [11].

Similarly, in the airways, changes in mucus induced by pollutants and abnormal RH generate conditions that favour infections or make their eradication more difficult [12].

In the face of epithelial damage, an excess of humidity in the indoor environment increases the allergen load of several allergens, such as moulds and mites, which, in turn, may facilitate the penetration of other allergens, such as *Staphylococcus* antigens [6]. Furthermore, higher levels of RH may increase the concentrations of some pollens and outdoor moulds, leading to an additional allergen load. On the other hand, dry air in the buildings is responsible for hoarseness in the throat and eyes, which is particularly relevant in patients with concomitant diseases (for example, Sicca syndrome).

## 3. RH Alteration and Its Direct Effects on Health

RH out of the normal range has been frequently reported in different settings and in different countries. In Northern Thailand, signs of high humidity were frequently observed in university buildings, such as condensation on windowpanes, water leakage, and indoor mould, which were significantly correlated with nasal and skin symptoms reported by the students’ community [13]. On the other hand, both conditions, moist and dry air, were responsible for an occupational risk of hoarseness in health care workers [14]. In a large study carried out in Norwegian private buildings, moisture or mould damage was observed in one out of three houses, with cooling rooms, basements, and bathrooms being the places at highest risk. Symptoms were also related to the years of construction, being more evident in older houses [15]. Interestingly, in a survey carried out in Korea among over 1000 subjects enrolled in a large office building, respiratory and extra-respiratory symptoms were more common in subjects with a longer stay. Moreover, after the refurbishment, though reduced, the symptoms did not disappear [16].

Exposure to low humidity causes sensory irritation symptoms in the eyes and airways, with a negative impact on work performance, sleep quality, virus persistence, and voice disruption [7]. In this condition, eyes are particularly at risk. In a large Chinese population, extremely low levels of atmospheric pressure andRH, as well as extreme levels of temperature, were significantly associated with an increased risk of outpatient visits for conjunctivitis. In the same study, high-speed wind was a protective factor [8]. The negative impact of dry air on the conjunctival mucosa is due to multiple mechanisms, such as the impairment of the lacrimal film due to water evaporation and the reduction of its thickness. Ocular symptoms include foreign body sensation, light hyperreactivity, blurred vision, and conjunctival redness [9]. Even the sensation of ocular dryness is strongly related to indoor humidity [17]. The worsening of eczema is related to RH, low indoor or high outdoor, due to different mechanisms, such as the reduction of skin conductance, the damage of the lipid layer, the activation of skin mastocytes and the reduction of elasticity [18]. In the long term, the thickness of the skin is reduced with the development of wrinkles. In general, with increasing RH, there was an improvement in the symptoms of all mucous membranes and the skin.

## 4. Interaction among Climate Change, Pollutants and RH

One of the effects of climate change is the increase in temperature and relative humidity outdoors and, consequently, indoors with more frequent overheating periods. Regarding pollutant-related factors, outdoor hygrothermal conditions can influence temperature and RH due to heat transfer through the building walls.

Outdoor-originated pollutants, such as polycyclic aromatic hydrocarbons (PAHs), benzene, and ozone, enter the indoor environment through the ventilation system, windows, and/or air infiltration through cracks [19].

However, ozone (O_3_) is essentially produced outdoors in the presence of ultraviolet light and precursors such as NOx. High concentrations of ozone are recorded during heat waves because high temperatures promote its formation. Studies have provided strong evidence that long-term exposure to high ozone concentrations increases mortality from respiratory and cardiovascular diseases, with adverse consequences for human health [20]. In order to break this evidence, a variety of attempts have been made to investigate the causes, starting with volatile organic compounds (VOCs), which are important precursors, to extensively alleviate O_3_ pollution. O_3_ and RH showed a non-linear relationship. When the RH was at low levels (RH < 22%), the vast majority of O_3_ concentrations remained at relatively low values, whereas as the RH increased, it was accompanied by higher O_3_ concentrations. With a further increase in RH, the O_3_ concentration decreased instead; a possible explanation is that too high RH could inhibit the reaction rate [21]. Fadeyi et al. showed that the rate of surface ozone deposition increases with increasing RH and temperature. This appeared more clearly at a humidity level higher than 50%. The ozone deposition velocity can increase by a factor of 17 when the humidity increases from 50% to 90%, depending on the type of material [22].

PM_2.5_ has been shown to have a positive relationship with temperature and concentrations and a negative relationship with wind speed due to the dilution effect. Precipitation, cloudiness, pressure, and RH showed a poor correlation with PM_2.5_ concentrations in most regions of the world. The increase in outdoor PM_2.5_ concentrations could be counterbalanced by the decrease in heating demand due to warmer winters and by the reduction of fossil fuel combustion. Lee et al. [23] studied PM_2.5_ infiltration, aiming to predict the relationship between future changes in outdoor temperature and fine particles by monitoring the indoor-outdoor sulphur ratio as an indicator of PM_2.5_ infiltration due to the absence of indoor sulphur sources. An increase in outdoor temperature of 2–3 °C in summer corresponds to up to a 0.06 increase in the indoor—outdoor sulphur ratio.

Human activities such as cooking, smoking, or window opening can influence indoor air quality. Changes in window opening can modify the air change rate and airflow velocities at indoor surfaces, which directly affect indoor pollutant concentrations. Window openings generally decrease in summer and increase in other seasons; this reduces ventilation rates and may increase exposure to radon and pollutants emitted from indoor sources such as materials, occupants, and combustion [24].

Different meteorological factors, such as changes in temperature, wind speed, air pressure, and humidity, can affect not only the concentration, distribution, and composition of particulate matter and gaseous pollutants, but also evaporation from the surface, thereby affecting the concentration and transmission of airborne allergens such as pollen [25]. The effects of climatic factors on neurodegenerative, cardiovascular, and some infectious diseases have been extensively studied [8].

Temperature variation can have a positive or negative correlation with hospital admissions or emergency room visits for asthma. This also suggests that weather variables, including RH, rainfall, thunderstorms, and wind, are factors that influence severe asthma exacerbations [26]. Furthermore, the combination of pollutants (CO, NO_2_, and SO_2_) and other indirect causal variables such as RH, wind speed, and temperature may also influence paediatric asthma visits [27].

## 5. Allergic Diseases and RH Influencing Effects

The epidemiologic association of asthma and rhinitis and the similarity of pathogenic mechanisms in the upper and lower airways represent the core of the ARIA (Allergic Rhinitis and Its Impact on Asthma) guidelines [28]. In fact, asthma, rhinitis, and nasal polyposis share the same inflammatory network, including epithelial damage, which can be the starting point of the inflammatory cascade. Two endotypes are identified in asthma as well as in nasal polyps: the eosinophilic (epidemiologically prevalent) and the neutrophilic endotype [28]. A crucial aspect is the epithelial release of alarmins (TLSP, IL33, and IL25) resulting in the activation of IL-C2 cells, which belong to innate immunity and drive eosinophil activation.

Though studies addressing the impact of RH alone on the respiratory epithelial barrier are lacking, elevated RH in combination with carbon nanoparticles has been shown to modify the microbiome and develop allergic asthma in animal models. Moreover, in the same study, this combination was responsible for bronchial hyperresponsiveness, airway inflammation, and remodelling [29]. Similarly, low RH induced epithelial corneal disruption in an animal [30]. Interestingly, in exercise-induced hyperpnea, an epithelial disruption has been demonstrated in all subjects and not only in asthmatic patients. In fact, an increase in CC16, an anti-inflammatory protein secreted by non-ciliated bronchial Clara cells, is observed in the urinary secretion. In cases of epithelial disruption, it passes passively into the bloodstream [31].

From a clinical point of view, although considered a “trivial disease”, allergic rhinitis represents the main risk factor for the development of asthma [32]. Furthermore, uncontrolled allergic rhinitis has a negative impact on concomitant asthma, being responsible for more admissions to emergency rooms, hospitalizations, and use of oral corticosteroids than in subjects without rhinitis. Therefore, among the preventive interventions for individuals with allergic asthma, the treatment of rhinitis, which is sometimes underestimated, is also crucial [33].

Moreover, among allergic diseases, a chronological progression from the skin to respiratory tract involvement can be identified, which has been called the “atopy march”. In this evolution, the first step is atopic dermatitis, where the disruption of the skin barrier promotes sensitization to both food and inhalant allergens [34]. In fact, the barrier damage due to genetically reduced filaggrin production and worsened by environmental factors, such as low RH, induces epithelial cells to release the above-mentioned alarmins, which in turn trigger a non-IgE-mediated eosinophilic inflammation. Moreover, the presence of pre-existing allergic inflammation with the release of IL-4, plays a role in a vicious circle, favouring further disruption of the epithelial barrier [35].

In allergic subjects, another trigger is the pollen load, often related to climatic factors and relative humidity. Caminati M. et al. evaluated emergency room admissions for asthma exacerbations and found that the most critical season was spring, when the presence of grass pollen in the Po Valley is at its highest peak [36]. In this regard, the impact of global warming, which has changed the pollen seasons by increasing both intensity and duration, should be emphasised [37].

Outdoor pollutants seem to have negative effects on asthma, favouring both a rapid functional decline and a higher frequency of exacerbations [38]. Moreover, a few years ago, fatal asthma cases were reported in young Italian allergic asthmatics during the peak in the atmosphere of Alternaria spores, to which they were sensitised [39]. Sensitization to mould and mites is related to indoor and outdoor humidity, and in a Chinese study, it was shown not only that sensitization has increased over the years but also the frequency of comorbid asthma and rhinitis [40]. In the same geographical region where this increase was detected, a progressive increase in external RHwas also detected.

High wind speeds can promote the dispersion of outdoor particulate matter (including pollen and mould spores), thus reducing its concentration. On the other hand, high temperatures and solar radiation can increase the effect of photosynthesis on ozone and thus affect the development of allergic diseases (e.g., eczema, allergic conjunctivitis) [41]. Gui SY et al. reported that low levels of RH increase the risk of outpatient visits for conjunctivitis. In fact, with a low level of RH(13%), the risk of outpatient visits for conjunctivitis increased by 12.3% compared to the number of visits required with a normal RH level (54%). A further increase in visits was registered at lower levels of RH. However, only a slight effect of HR was shown in the multiple meteorological factor model (RR = 1.021–1.025) [8]. Furthermore, the study results suggested that a low RH was significantly associated with the risk of conjunctivitis during the warm season. However, in another group of patients, low RH and air pressure were found to be risk factors for conjunctivitis in summer as well as in winter, whereas wind speed proved to be a protective factor, which may be related to the fact that elevated temperatures and high humidity provide a suitable environment for bacteria and allergens to live. Weather conditions are known to induce and exacerbate allergic diseases such as allergic conjunctivitis, through the interplay of pollen and air pollutants.

Mites are common domestic allergens in atopic asthma patients [42]. Another indoor risk factor is the presence of moulds such as *Cladosporium and Aspergillus*. Moses L. et al. identified a significant prevalence of respiratory symptoms in mould-sensitised patients in a population of elderly, mostly female, subjects [43]. This finding has clinical relevance, as the elderly spend most of their lives indoors. There is also a significant relationship between the concentration of mould in the microenvironment and the production of allergens (proteases) by mites, which can degrade fungal walls [44]. Mites are also very susceptible to environmental conditions (RH, temperature, and air pollution). There is evidence of an increase of mite growth in subtropical and tropical areas, though the reasons are still unclear [44]. The influence of storage conditions on *Tyrophagus putrescentiae* (TP) infestation, related to the prevalence of mite hypersensitivity, has been reported in Taiwanese workers sensitised to food storage mites [45]. In this study, humidity is the most important growth factor for TP. Identification of TP in drugstores is more common than in ordinary houses. This storage mite has been mainly identified in pet food and mushrooms. A possible relationship between the presence of bacteria and fungi in the indoor environment and the microbiome and mycobiome of the airways of subjects with severe asthma was investigated. During periods of stabilisation of the disease, a greater variety of fungi was detected; however, during asthma exacerbations, the number of varieties of mycophytes increased both in the indoor environment and consequently in the airways [46].

## 6. Infections and RH Influencing Effects

It has been stated that dry air and excessive humidity can both negatively affect health conditions [1,47]. Lowen AC et al., in a study on animal models, showed that both low temperature and reduced RH were risk factors for the spread of the influenza virus [5]. Breathing dry air, alteration of the nasal mucosa, epithelial damage, and reduced mucociliary clearance was observed. The stability of influenza virus in an aerosol was highest at low RH (20–40%), lowest at intermediate RH (50%), and acceptable at low RH (60–80%). Key determinants in the transmission of the influenza virus are the “droplet nuclei”. In fact, at low RH, the evaporation of water from the exhaled bioaerosol occurs rapidly; at high RH, small respiratory droplets absorb water, increase in size, and remain for longer in the air. Viral infectivity is highest at low RH and gradually decreases by increasing the RH to different dimensions [48]. In a Chinese multicenter study, it has been reported that meteorological factors contribute to the risk of pulmonary tuberculosis, and RH over 80% is negatively associated with the risk of infection [49]. Similar results are obtained regarding SARS-CoV-2 virus infection, where low outdoor humidity is associated with COVID-19 outbreaks [50].

The increase in the frequency of extreme precipitation events (EPEs) has been the subject of studies [51]. Extreme precipitation leading to flooding also causes damage to many aspects of the built environment and also modifies human behaviour. During EPEs, people change their behaviour and prefer to stay indoors rather than spend time outdoors or in public, increasing person-to-person contact frequency. Solar as well as artificial UV irradiation has been shown to inactivate many microbes, including viruses such as influenza, RSV, and the coronavirus responsible for COVID-19 [51,52]. The alterations of the natural environment that occur during EPEs are related to the increase in RH, which continues to rise following precipitation due to evaporation. Furthermore, during EPEs, cloud cover reduces ultraviolet (UV) irradiation, lengthening the environmental survival time of infectious viruses. The discomfort outside the home leads to a higher demand for the warming and dehumidification effects of indoor heating, ventilation, and air conditioning (HVAC) systems. When weather events drive populations indoors, the indoor built environment can play a significant role in fostering or hindering the transmission of infectious diseases [52]. The respirability of infectious droplets depends on a balance between droplet size and evaporation rate. HVAC systems may increase the likelihood of transmission by removing humidity from indoor air, lowering indoor vapour pressure, and thus slowing down the evaporation of droplets.

## 7. Unhealthy Environment and Interventions

The perception of an unhealthy environment is strongly conditioned by increasing or decreasing RH; unpleasant conditions and pungent odours are due to mould overgrowth [53], which is related to increased RH and is more frequent in older houses [15,16]. Damage to buildings due to humidity seems to be related to respiratory and non-respiratory symptoms, especially in subjects with prolonged stays, and the symptoms persist even if corrective structural interventions have been carried out [16]. Surprisingly, high RH has been identified as a risk factor for suicide. In particular, females and younger age groups seem to be more affected by high RH and heatwaves [54]. RH below or above the recommended levels may also affect the storage stability of drugs, particularly those administered by inhalation [55]. A limited number of studies underline the positive role of humidity reduction, especially in improving work activity and reducing school absenteeism following indoor air humidification [56]. Standard ventilation, in a few cases, was more efficient than mechanical ventilation, allowing optimal humidity levels [47]. Moreover, the use of a ventilation system resulted in a significant reduction in pathogens, reducing the life expectancy of the influenza virus [57]. Intervention on indoor temperature and RH using control devices determines a significant decrease in internal exposure to dust mites and a consequent decrease in allergic symptoms in children with allergic rhinitis [58].

## 8. Conclusions

Climate change and pollution are two of the most significant problems affecting our daily lives. The impact of these issues is felt in a variety of ways, from changes in weather patterns to air and water pollution. The challenge of creating healthier environments, both indoors and outdoors, has become one of the most pressing issues of our time. However, this challenge is complicated by the complex and interconnected nature of these problems. With so many pollutants and climate factors in play, it can be difficult to determine the exact role of each cause. Moreover, though most pollutant have an impact on the respiratory mucosa, they have different mechanisms of action and impact contemporaneously. Additionally, the outdoor environment can be monitored fairly easily, especially in industrialised countries, and can affect a large population, whereas the level of indoor pollution can be different in different buildings and should only be monitored regularly in public buildings. RH is an undervalued meteorological factor that can have an impact on both the outdoor and indoor environment and can negatively affect it through several mechanisms. In fact, damage to the integrity of the mucous membranes and skin barrier can increase the risk of allergic sensitization and, at the same time, trigger the Th-2 inflammatory cascade. The disruption of the epithelial barrier is the consequence of the alteration of the relative humidity in the epithelial layer and the consequent impairment of the mucociliary clearance. Higher or lower levels of relative humidity can adversely affect the eyes, skin, and respiratory tract. High RH may increase allergen loads such as pollen, mites, and mould. To note, fungi sensitization (alternaria and cladosporium) is related to fatal asthma events and has a possible negative effect on the health of the elderly.

In conclusion, this review has summarised the current evidence regarding the role of RH in the immune system, infections, respiratory health, and allergies. While often overlooked, RH emerges as a significant factor that warrants attention in prevention strategies. In fact, the control of relative humidity results in a strategy for prevention and control of the spread of viruses and bacteria. Tools for controlling relative humidity can play a preventive role in reducing the risk of infection and improving respiratory and skin diseases. Though today great attention is paid to tools that can purify the indoor environment, careful control of RH and temperature can be carried out regularly in order to obtain healthier places to live and work.

Further research is needed to better understand the specific mechanisms by which RH influences immune responses and to explore its potential as a target for intervention. By incorporating RH into comprehensive health assessments and public health policies, we can take a proactive approach towards improving respiratory health and enhancing overall well-being.

## Figures and Tables

**Figure 1 ijms-24-09456-f001:**
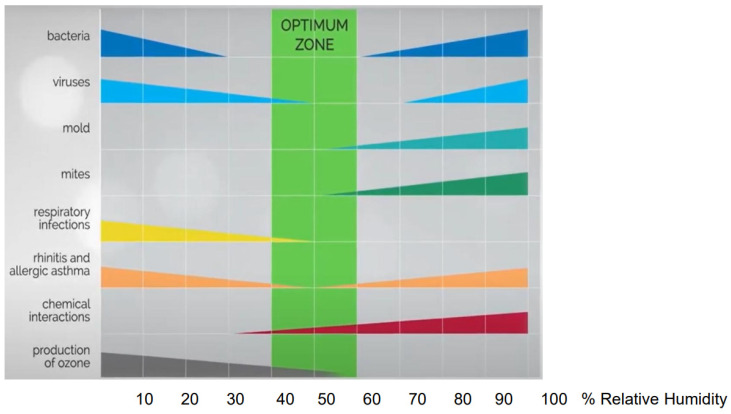
Optimal Relative Humidity Ranges for Health. A decrease in bar width indicates a decrease in effect (modified from Sterling EM et al., 1985 [1].

**Figure 2 ijms-24-09456-f002:**
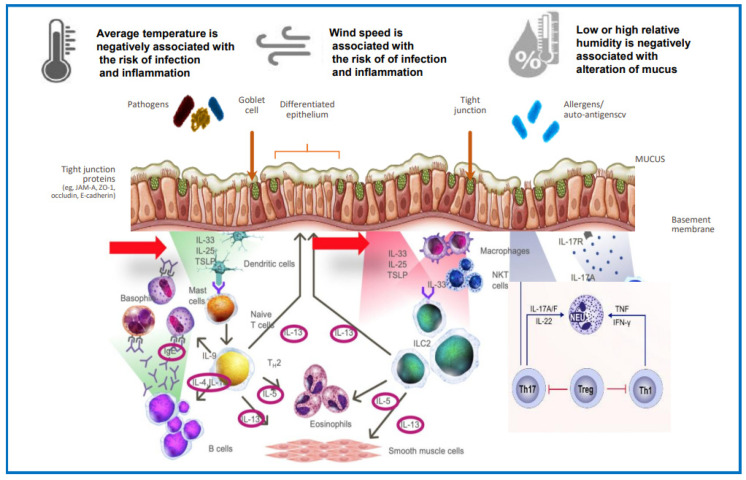
Meteorogical factors contribute to the relative humidity effect on the rheologic properties of mucociliary clearance. The alteration of the integrity of the physical barrier maintained by mucus and tight junctions increases the susceptibility to pollutants, allergens, and pathogens.

**Table 1 ijms-24-09456-t001:** Summary of the main mechanisms by which RH determines its effects on human health.

Relative Humidity Effects	Mechanisms	Citations
Increased infections	- Increased bacterial growth (RH < 30% or > 60%) - Increased viral growth, stability and infectivity (RH < 50% or > 70%) - Relevant prevalence of indoor fungi in the home air when RH > 60%.	[1,5]
Increased allergen burden	Excess humidity in the indoor environment is associated with the growth of allergens, such as moulds and mites.	[6]
Reduced mucociliary clearance	- RH affects mucus osmolarity. - In the lower airways, excess mucus or its increased thickness may favour bacterial engraftment.	[2,3,4]
Mucosal impairment	- Alteration of the mucous layer can lead to a disruption of epithelial junctions, increasing permeability to pollutants, allergens and viruses. - Damaged epithelial cells may release mediators, such as alarmins (IL33, TSLP, IL25), which can promote the inflammatory T2 cascade. - T2 cytokines, such as IL-4 and IL-13, downregulate the expression of tight junction barrier proteins, contributing to reduce epithelial junction integrity and increase barrier permeability in a vicious cycle. - Dry air on the conjunctival mucosa can lead to impairment of the lacrimal film due to evaporation of water and reduction of its thickness. - Alterations in RH are related to worsening eczema, due to reduced skin conductance, damage to the lipid layer, activation of skin mast cells and reduced elasticity.	[2,3,4,7,8,9]

## Data Availability

All the data are presented in this study.

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
