# Peer review of "Relative Humidity and Its Impact on the Immune System and Infections"

_ijms, 2023, doi:10.3390/ijms24119456_

Round 1
Reviewer 1 Report
The manuscript conducts a review of the role of Relative Humidity (RH) on health effect such as respiratory diseases (asthma, COPD, ..) and the transmission of virus, bacteria, mites, pollens,..). Basically the if the RH is too low <40% or too high > 60%, the respiratory system does not cope well and various different types of symptoms and health effects occur as a result. I am not clear why the range 40% < RH < 60% is an optimum range. The authors list out a lot of studies related to the environment factors, pathogens and the physological responses of the respiratory system that lead to the health outcomes. But the central message is not clear about the aim and findings of this study. What are the main findings and the authors should clearly list out these findings.
Other comments
(1) Throughout the manuscript, RH and HR are used interchangely. Should use RH (Relative Humidity) rather HR.
(2) Line 34: "conflicting effects of an increase or decrease in relative humidity". Do you mean "conflicting health effects" ?
(3) Line 68: COPD should specify the full name here as it first appears.
(4) Line 105: "... may vehicle other allergens". I dont understand this sentence.
(5) Line 185-186: "The effects of climatic factors on neurodegenerative, cardiovascular and some infectious diseases have been extensively studied". Please give references for this statement.
(6) Line 195: "ARIA". Please give a full name for this
English is fine. Only minor change is required.
Author Response
Dear Review 1
below we have given the point-by-point responses to your comments:
-The manuscript conducts a review of the role of Relative Humidity (RH) on health effect such as respiratory diseases (asthma, COPD, ..) and the transmission of virus, bacteria, mites, pollens,..). Basically the if the RH is too low <40% or too high > 60%, the respiratory system does not cope well and various different types of symptoms and health effects occur as a result. I am not clear why the range 40% < RH < 60% is an optimum range. The authors list out a lot of studies related to the environment factors, pathogens and the physological responses of the respiratory system that lead to the health outcomes. But the central message is not clear about the aim and findings of this study. What are the main findings and the authors should clearly list out these findings.
Thank you for the good comment and suggestion. We have improved the text and made the objectives and findings of the study clearer
Other comments
(1) Throughout the manuscript, RH and HR are used interchangely. Should use RH (Relative Humidity) rather HR. Thank you, done. See the text please
(2) Line 34: "conflicting effects of an increase or decrease in relative humidity". Do you mean "conflicting health effects" ? Yes, thank you. See the text please
(3) Line 68: COPD should specify the full name here as it first appears. Done, please see the text
(4) Line 105: "... may vehicle other allergens". I dont understand this sentence.
Higher humidity in the indoor environment induces an epitelial damage, that facilitates the increases allergen load, such as molds and mites, which, in turn, may facilitates the penetration of other allergens, as the staphylococcus antigens
(5) Line 185-186: "The effects of climatic factors on neurodegenerative, cardiovascular and some infectious diseases have been extensively studied". Please give references for this statement. Thank you, done. See the text please
(6) Line 195: "ARIA". Please give a full name for this. Done, please see the text
Comments on the Quality of English Language
English is fine. Only minor change is required. Done, thanks.
Reviewer 2 Report
General comments:
The topic of the review is very current and the authors considered several important factors related to relative humidity.
Several factors were discussed in relation to infections, allergies and other pathologies.
Environmental factors as well as climate change are take into account.
Specific comments:
Lines 40,41, 105, 260: names of species in Italics
Table 1: delete yellow
Lines 141, 334: outdoor and indoor (no outdoors and indoors)
Line 190: NO2, SO2) write NO2, SO2
References: write as reported by Instrucion to the authors
Author Response
Dear Reviewer 2
below we have given the point-by-point responses to your comments:
General comments: The topic of the review is very current and the authors considered several important factors related to relative humidity. Several factors were discussed in relation to infections, allergies and other pathologies. Environmental factors as well as climate change are take into account.
Specific comments:
Lines 40,41, 105, 260: names of species in Italics Yes, thank you. See the text please
Table 1: delete yellow Yes, thank you. See the table.
Lines 141, 334: outdoor and indoor (no outdoors and indoors) Yes, thank you. See the text please
Line 190: NO2, SO2) write NO2, SO2. Done
References: write as reported by Instrucion to the authors. Thank you. We revised all references as reported by Instruction to the authors. See the text please
Round 2
Reviewer 1 Report
The authors have adequately addressed my comments.
Please change HR to RH throught the text